# PERFECTED enhanced recovery pathway (PERFECT-ER) versus standard acute hospital care for people after hip fracture surgery who have cognitive impairment: a feasibility cluster randomised controlled trial

Jane L Cross ![ORCID],[1] Simon P Hammond ![ORCID],[2] Lee Shepstone,[3] Fiona Poland,[1] Catherine Henderson,[4] Tamara Backhouse ![ORCID],[1] Bridget Penhale,[1] Simon Donell,[5] Martin Knapp,[4] Douglas Lewins,[1] Alasdair MacLullich,[6] Martyn Patel,[5] Opinder Sahota ![ORCID],[7] Toby O Smith,[1] Justin Waring,[8] Robert Howard,[9] Clive Ballard,[10] Chris Fox[10]

**Correspondence to**
Dr Jane L Cross;
j.cross@uea.ac.uk

## ABSTRACT

**Objectives** Assess feasibility of a cluster randomised controlled trial (RCT) to measure clinical and cost-effectiveness of an enhanced recovery pathway for people with hip fracture and cognitive impairment (CI).

**Design** Feasibility trial undertaken between 2016 and 2018.

**Setting** Eleven acute hospitals from three UK regions.

**Participants** 284 participants (208 female:69 male). Inclusion criteria: aged >60 years, confirmed proximal hip fracture requiring surgical fixation and CI; preoperative AMTS ≤8 in England or a 4AT score ≥1 in Scotland; minimum of 5 days on study ward; a 'suitable informant' able to provide proxy measures, recruited within 7 days of hip fracture surgery. Exclusion criteria: no hip surgery; not expected to survive beyond 4 weeks; already enrolled in a clinical trial.

**Intervention** PERFECT-ER, an enhanced recovery pathway with 15 quality targets supported by a checklist and manual, a service improvement lead a process lead and implemented using a plan–do–study–act model.

**Primary and secondary outcome measures** Feasibility outcomes: recruitment and attrition, intervention acceptability, completion of participant reported outcome measures, preliminary estimates of potential effectiveness using mortality, EQ-5D-5L, economic and clinical outcome scores.

**Results** 282 participants were consented and recruited (132, intervention) from a target of 400. Mean recruitment rates were the same in intervention and control sites, (range: 1.2 and 2.7 participants/month). Retention was 230 (86%) at 1 month and 54%(144) at 6 months. At 3 months a relatively small effect (one quarter of an SD) was observed on health-related quality of life of the patient measured with EQ-5D-5L proxy in the intervention group.

**Conclusion** This trial design was feasible with modifications to recruitment. Mechanisms for delivering consistency in the PERFECT-ER intervention and

## Strengths and limitations of this study

► This feasibility randomised controlled trial provides valuable evidence that the intervention and trial design can be delivered but would require a substantially larger number of trial sites and larger sample size.

► As only a small proportion of people of non-white ethnicity were recruited (patients and suitable informants) it is unclear how successful recruitment and retention of participants from wider ethnic backgrounds would be.

► The duration and type of cognitive impairment, that is, established dementia versus temporary delirium, was not controlled for within the analysis.

► Health economic data collection should be simplified and data extracted from hospital records to reduce burden on suitable informants.

participant retention need to be addressed. However, an RCT may be a suboptimal research design to evaluate this intervention due to the complexity of caring for people with CI after hip fracture.

**Trial registration number** ISRCTN99336264.

## INTRODUCTION

Hip fracture is associated with advancing frailty and has substantial impact on the health, well-being and independence of older people and their families.[1 2] Acute hip fracture care costs an estimated £1.1 billion per annum in the UK.[3] In the 12 months after fracture, patients are at increased risk of cognitive and functional decline, admission to long-term care institutions and higher

mortality.[4] People with cognitive impairment (CI) are among the most vulnerable in acute hospital settings,[5] with lower short-term survival and 15% mortality during admission.[4] They are susceptible to suboptimal and inconsistent care standards that contribute to cognitive deterioration, increase risk of postoperative complications, prolong length of stay and cause loss of independence.[6]

In older adults with hip fracture, approximately 19% have dementia and up to 42% some degree of CI that may not meet criteria for a dementia diagnosis.[7] People with hip fracture and CI are frequently cared for in environments which deliver excellent hip fracture care but are less skilled managing people with CI.[8 9] Hospital care of patients with CI remains an ongoing area of concern[5] with systemic failures in the care of older people repeatedly identified.[10] Hospital staff may lack the knowledge and skills necessary to identify and assess CI, leading to underidentification which negatively affects access to rehabilitation services, supported discharge planning, person-centred care plans and involvement of families and carers.[11–14]

This study assessed the feasibility of a cluster design randomised controlled trial (RCT) to measure the clinical and cost-effectiveness of an enhanced recovery pathway versus standard care in acute hospitals for people after hip fracture surgery who demonstrate CI. Feasibility objectives included recruitment, retention, outcome selection, sample size estimation and acceptability of intervention training and delivery in National Health Service (NHS) services.

## METHODS
This paper has been prepared in accordance with the Consolidated Standards of Reporting Trials (CONSORT) Extension for Pilot and Feasibility Studies[15] reporting guideline. The study methods are summarised below and previously reported in detail.[16]

### Public and patient involvement
Patients and the pubic were involved from the conception of this study, through the review and funding process, the study, analysis and writing the findings. They were part of the steering, oversight and data monitoring groups.

### Design and setting
A multicentre, feasibility, cluster RCT was undertaken between 2016 and 2018. In line with MRC guidance for complex interventions, an integrated process evaluation was conducted[17]; this is currently under review.

### Randomisation
Randomisation was stratified by geographical area, with one intervention and one control hospital in UK region. Ten NHS hospitals were randomised to deliver experimental (PERFECT-ER) or control interventions. An additional site was recruited as a control group in July 2017 when another control site failed to recruit, and

recruitment was extended from 10 months to 15 due to difficulties recruiting suitable informants. Recruitment was between November 2016 to February 2018.

### Participants
#### Inclusion criteria
Participants were included if:
► Confirmed proximal hip fracture requiring surgery.
► Aged 60 years or over at the time of surgery.
► Preoperative Abbreviated Mental Test Score (AMTS ≤8 in England (including those with zero because of an inability to answer questions) or a 4AT score ≥1 in Scotland.
► Minimum of 5 days on the study ward.
► Patient had a 'suitable informant' (eg, relative, unpaid or paid carer, care home manager) with a minimum of once a month face-to-face or telephone contact with the patient and able to provide proxy measures where required.
► Both patient and suitable informant to be recruited into the trial within 7 days of the hip fracture surgery.

#### Exclusion criteria
Participants were excluded if:
► Did not undergo hip surgery.
► Patient not expected to survive beyond 4 weeks.
► Patient already enrolled in a clinical trial of an investigational medicinal product.

### Sample size
The target sample was 400 patient participants (200 per arm) from 10 centres (40 patient participants per site), based on the degree of precision for the estimated intraclass correlation coefficients (ICC). This was expected to provide a SE for the ICC of between 0.033 and 0.041, for a true ICC value of between 0.05 and 0.10 for any endpoint. A priori, it was expected that four participants would be recruited per site, per month, over 10 months recruitment period.

### Participant recruitment and consent
A three-step recruitment process was implemented, guided by previous phases of the PERFECTED programme, previous studies[18 19] and input from clinical and academic collaborators:
1. Research nurses identified all new hip fracture admissions and screened for prerecruitment eligibility in collaboration with clinical staff.
2. Patients (and where possible their potential suitable informant) were approached by the research nurse who provided study information as soon as clinically appropriate. Mental capacity was assessed by the research nurse, according to the appropriate legislative frameworks. In those lacking capacity to consent, consultee agreement from a relative or professional caregiver was sought, following the requirements of UK capacity legislation.[20 21]
3. The research nurse approached the patient and suitable informant to obtain written informed consent.

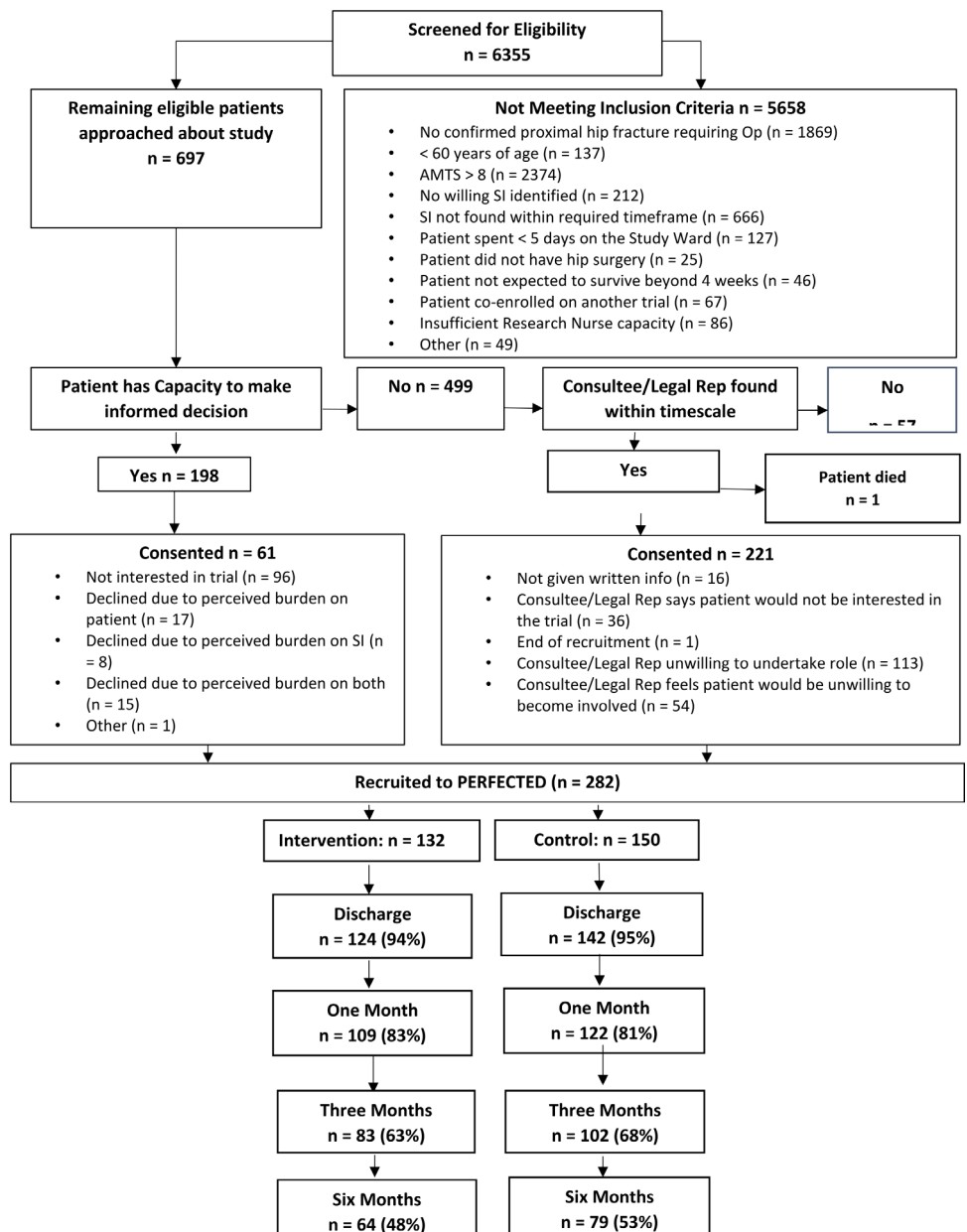

**Figure 1** Patient flow diagram. SI; Significant Informat, AMTS; Abbreviated Mental Test Score.

## Intervention

### Experimental intervention: PERFECT-ER

The PERFECT-ER is a multicomponent intervention, implemented using service improvement principles, comprising:

▶ The PERFECT-ER checklist and manual.
▶ A Service improvement lead (SIL) and PERFECTED process lead (PPL).
▶ A model for change (plan–do–study–act).[22]

The checklist has 15 organisational items, and 68 individual patient items grouped into three stages (admission and preoperative; postoperative and rehabilitation; and discharge), reflecting the patient journey through acute care settings. It was designed to identify areas of strength, and potential for improvement in practice, and overarches current hip fracture guidance. A comprehensive

handbook explaining how to implement and use the intervention (the PERFECT-ER manual) was provided.

In the 3 months prior to recruitment commencing, the intervention was implemented in intervention sites by the SIL working 0.50 FTE, following the handbook and adherence assessed. When sites commenced recruitment, SIL resource was reduced to 0.2 FTE for the study period. A senior clinician (PPL) assisted the SILs for an hour a week to implement PERFECT-ER then an hour per month during recruitment.

### Comparator group

The control group received treatment as usual. What this consisted of was recorded to determine local practice which followed National Institute for Clinical Excellence (NICE) guidance for hip fracture care[23]

**Table 1** Recruitment rates by centre

| Group | Site | Start date | Months | Recruited | Rate/month |
|---|---|---|---|---|---|
| **Intervention** | | | 70 | 132 | 1.9 |
| | 01 | December 2016 | 14 | 26 | 1.9 |
| | 03 | November 2016 | 15 | 34 | 2.3 |
| | 06 | November 2016 | 15 | 30 | 2.0 |
| | 07 | February 2017 | 12 | 19 | 1.6 |
| | 10 | December 2016 | 14 | 23 | 1.6 |
| **Control** | | | 81 | 150 | 1.9 |
| | 02 | November 2016 | 15 | 24 | 1.6 |
| | 04 | November 2016 | 15 | 18 | 1.2 |
| | 05 | November 2016 | 15 | 23 | 1.5 |
| | 08 | November 2016 | 15 | 35 | 2.3 |
| | 09 | November 2016 | 15 | 40 | 2.7 |
| | 50 | July 2017 | 6 | 10 | 1.7 |
| **Total** | | | 151 | 282 | 1.87 |

## Outcomes

Data were collected from medical records of participating hospitals, the National Hip Fracture Database (NHFD)[24] and participants and suitable informants (summarised in online supplemental table 1). Study feasibility outcome measures included: recruitment and attrition, intervention acceptability and fidelity, completion of participant reported outcome measures. The delivery of the intervention was monitored by auditing the patients notes against the PERFECT-ER checklist. Five patients per site were audited at the beginning of each implementation cycle and at the end of the trial: at 3 months pretrial, 1.5 months pretrial, trial baseline, 4 months, 7 months, 10 months, 13 months and 15 months. Clinical outcomes: mortality rate at 30 and 120 days; Bristol Activities of Daily Living Scale; hospital admissions (number, length of stay and time to first admission); falls and mortality during previous 6 months and the number of medications. Economic measures: quality-adjusted life-years (QALY) of the participant[1] computed from DEMQOL-U and DEMQOL-PROXY-U) and[2] computed from EQ-5D-5L completed by participants and again by proxy, QALY of the suitable informant (unpaid carer), use of health, social and unpaid care collected via the client services receipt inventory (CSRI)[25] and hospital service use abstracted from hospital records. Costs of the intervention were assembled from time inputs of personnel providing PERFECT-ER, including time spent championing the ERP in study setup (online supplemental table 2). Costs of inputs per site were calculated by dividing the costs of each role by the number of potentially affected patients on each study ward over the intervention period. Unit costs for other services were from published sources.[26–29]

## Statistical analysis
### Clinical outcome analysis
The data analyses summarise study process information including recruitment, participant 'flow' and retention,

sample characteristics and completeness of baseline and follow-up outcome measures. To assess fidelity of the intervention the mean 'PERFECT-ER' score of enacted checklist items was determined.

For each outcome measure, at each follow-up point, an ICC was calculated together with a 95% CIs. These were calculated to assist the choice of primary outcome measure and inform potential sample size calculations for a definitive trial.

A precise estimate of intervention efficacy was not a primary objective of the data analyses. However, all efficacy outcome measures were modelled using a general linear model including the baseline value of the outcome (where available) and the treatment arm. Generalised estimating equations were used to account for 'clustering' created by the hospital-level randomisation, thus accounting for the lack of independence of patient-level data within individual hospitals. The estimates of between arm difference are provided with 95% CIs. The relationship between the individual 'PERFECT-ER' score and outcomes was considered and a Pearson correlation coefficient calculated to assess the strength of the linear relationship. The difference in mean 'PERFECT-ER' score between those known to have died during the study and those known to have survived was also calculated.

### Economic analysis
The economic evaluation took an NHS and Personal Social Services (social care) perspective and a societal perspective, incorporating costs of unpaid care and out-of-pocket expenses (for equipment, adaptations, travel to healthcare appointments).

We computed utilities (to subsequently calculate QALYs) using societal weights (DEMQOL-U from the DEMQOL; DEMQOL-Proxy-U from the DEMQOL-Proxy; and EuroQuo 5D 5L (EQ-5D-5L).[30 31] QALYs over the intervention period were derived using the trapezoid method to approximate the area under the quality of life curve, with linear interpolation between time points.

We examined the ICC of QALY and total costs at 6-month follow-up, with Searle's confidence intervals (using the arithmetic mean cluster size for unbalanced data) derived from one-way analysis of variance.[32]

We examined the extent to which hospital services use extracted from hospital records gave the same estimates as data collected by suitable informant report. We examined the level of agreement on frequency of service use (counts) and total hospital costs between the two sources as estimated by Lin's concordance correlation coefficient.[33] We also examined agreement between sources using the 95% limits of agreement approach,[34] which calculates means and SD of paired differences and the CI for the difference, conditional on those differences being normally distributed and independent of the measures' magnitudes.[35] Research nurses recorded the time taken to complete sections of the PERFECT-ER case report forms, covering multiple instruments/questions. To calculate a time-per-question estimate, the time taken to complete

**Table 2** Participant and suitable informant baseline characteristics

| Participant characteristic | Intervention (N=132) | Control (N=150) | Total (N=282) |
|---|---|---|---|
| Consent: | | | |
| Providing own consent | 23 (17.6%) | 38 (25.9%) | 61 (21.9%) |
| Consultee/legal rep consent | 109 (82.4%) | 112 (74.1%) | 221 (78.1%) |
| Age (mean (SD)) | 85.5 (7.4) | 86.4 (7.9) | 86.0 (7.6) |
| Missing | 2 | 3 | 5 |
| Gender: | | | |
| Male | 37 (28.0%) | 32 (22.1%) | 69 (24.9%) |
| Female | 95 (72.0%) | 113 (77.9%) | 208 (75.1%) |
| Missing | 0 | 5 | 5 |
| Ethnicity: | | | |
| Asian | 1 (0.8%) | 5 (3.4%) | 6 (2.2%) |
| Black | 1 (0.8%) | 0 | 1 (0.4%) |
| White | 106 (80.9%) | 118 (80.8%) | 224 (80.9%) |
| Unable to respond | 23 (17.6%) | 23 (15.8%) | 46 (16.6%) |
| Missing | 1 | 4 | 5 |
| Status: | | | |
| Married/partner | 40 (30.5%) | 48 (32.7%) | 88 (31.7%) |
| Divorced | 7 (5.3%) | 8 (5.4%) | 15 (5.4%) |
| Single | 6 (4.6%) | 4 (2.7%) | 10 (3.6%) |
| Widowed | 54 (41.2%) | 60 (40.8%) | 114 (41.0%) |
| Unable to respond | 24 (18.3%) | 27 (18.4%) | 51 (18.3%) |
| Missing | 1 | 3 | 4 |
| Employment status: | | | |
| Employed | 3 (2.3%) | 3 (2.1%) | 6 (2.2%) |
| Unemployed | 3 (2.3%) | 3 (2.1%) | 6 (2.2%) |
| Retired | 98 (74.8%) | 107 (73.3%) | 205 (74.0%) |
| Unable to respond | 27 (20.6%) | 33 (22.6%) | 60 (21.7%) |
| Missing | 1 | 4 | 5 |
| **Suitable informant characteristic** | **Intervention (N=132)** | **Control (N=150)** | **Total (N=282)** |
| Contact: | | | |
| Face to face | 121 (91.7%) | 129 (90.8%) | 250 (91.2%) |
| Phone call | 8 (6.1%) | 11 (7.7%) | 19 (6.9%) |
| Postal | 3 (2.3%) | 2 (1.4%) | 5 (1.8%) |
| Missing | 0 | 8 | 8 |
| Relationship: | | | |
| Spouse | 26 (19.8%) | 26 (18.3%) | 52 (19.0%) |
| Other family member | 98 (74.8%) | 110 (77.5%) | 208 (76.2%) |
| Non-family member | 4 (3.1%) | 4 (2.8%) | 8 (2.9%) |
| Paid carer | 3 (2.3%) | 2 (1.4%) | 5 (1.8%) |
| Missing | 1 | 8 | 9 |
| Age (mean (SD)) | 60.7 (13.1) | 62.2 (12.6) | 61.5 (12.9) |
| Missing | 4 | 10 | 14 |
| Gender: | | | |
| Male | 46 (34.8%) | 63 (44.4%) | 109 (39.8%) |
| Female | 86 (65.2%) | 79 (55.6%) | 165 (60.2%) |

Continued

| Participant characteristic | Intervention (N=132) | Control (N=150) | Total (N=282) |
|---|---|---|---|
| Missing | 0 | 8 | 8 |
| **Ethnicity:** | | | |
| *Asian* | 1 (0.8%) | 7 (4.9%) | 8 (2.9%) |
| Black | 2 (1.5%) | 0 | 2 (0.7%) |
| White | 129 (97.7%) | 135 (95.1%) | 264 (96.4%) |
| Missing | 0 | 8 | 8 |
| **Status:** | | | |
| Married/partner | 98 (77.2%) | 109 (77.3%) | 207 (77.2%) |
| Divorced | 11 (8.7%) | 8 (5.7%) | 19 (7.1%) |
| Single | 15 (11.8%) | 16 (11.3%) | 31 (11.6%) |
| Widowed | 3 (2.4%) | 8 (5.7%) | 11 (4.1%) |
| Missing | 5 | 9 | 14 |
| **Employment status:** | | | |
| Employed | 63 (48.1%) | 54 (38.0%) | 117 (42.9%) |
| Unemployed | 11 (8.4%) | 21 (14.8%) | 32 (11.7%) |
| Retired | 57 (43.5%) | 67 (47.2%) | 124 (45.4%) |
| Missing | 1 | 8 | 9 |

**Table 2** Continued

the CSRI, hospital use and medications review questions was divided by the number of items in the respective sections. Time taken to complete the measures was calculated by multiplying the total number of questions by the time-per-question.

Indicative cost-effectiveness analyses were conducted but are not reported here; details are available from the corresponding author.

## RESULTS

### Participant recruitment and retention

Figure 1 illustrates patient flow. Recruitment rate by centre is presented in table 1. Hospital characteristics at baseline are described in online supplemental table 3, which shows sites in both intervention and control groups are broadly similar. 282 participants, 132 from intervention sites and 150 from control, were recruited. There were 151 months of site recruitment, 70 in intervention and 81 in control sites. Average recruitment rates did not differ between intervention and control sites, ranging from 1.2 to 2.7 participants/month. Mean recruitment rate was 1.87 per site/month. This contrasts with the expected four per site/month. The demographic characteristics of the 282 study participants and suitable informant characteristics are shown in table 2.

Overall, the attrition rate was 50.7% (143/282). For the PERFECT-ER intervention attrition was 48.5% (64/132) and for control 52.7% (79/150).

### Intervention delivery

Although implementation was standardised across sites overall compliance with the intervention fluctuated over time and between sites. This is explored fully in the process evaluation (under review).

### Missing data

The degree of missing data varied across measures and across time points. For example, baseline data collection consistently demonstrated high missingness for all outcomes (online supplemental table 4). In contract, at discharge onwards, there were low missingness with the exception of the HowRwe at discharge EQ-5D-5L. Patient at 1, 3 and 6 months, and the Timed Up and Go at 3 months. The EQ-5D-5L for the suitable informant and proxy both demonstrated high missingness at 6 months in the intervention group (online supplemental table 4).

### Economic outcomes

For economic data collection, there was relatively low occurrence of missing data for all health utilisation variables in primary care (6%–8%) and hospital care, including both suitable informant-reported and hospital records-extracted use of emergency department, inpatient and outpatient services (4%–13%). Of a maximum of 23 medications reported, 3–4 costs were missing per case across the time points. More data were missing for suitable informant-reported unpaid care and lost working time. This was primarily because research nurses did not indicate whether the suitable informant was an unpaid or paid carer in 25% of cases at baseline and 17%, 15% and 13% of cases at 1, 3 and 6 months follow-up, respectively. Where the suitable informant was identified as an unpaid carer, rates of missingness in the unpaid carer questions were between 2% and 8% at the first three time points and 2%–11% at 6-month follow-up.

**Table 3** Estimates of outcome

| Time point and outcome measure | Intervention (N=132) Mean (SD) | Control (N=150) Mean (SD) | Adjusted difference[*] | 95% CI | P value |
|---|---|---|---|---|---|
| **Baseline** HowRThey | 4.96 (2.87) | 4.55 (3.20) | | | |
| HowRwe | 8.76 (2.38) | 9.11 (2.23) | | | |
| EQ-5D—patient | 0.24 (0.37) | 0.32 (0.36) | | | |
| **EQ-5D—SI** | 0.80 (0.24) | 0.85 (0.23) | | | |
| **EQ-5D—proxy** | −0.01 (0.23) | 0.15 (0.33) | | | |
| MMSE | 12.2 (8.0) | 10.8 (8.8) | | | |
| BADLS | 24.3 (14.0) | 21.0 (14.7) | | | |
| 4AT | 4.02 (3.33) | 4.80 (4.02) | | | |
| CDR | 1.63 (0.98) | 1.41 (0.95) | | | |
| **Discharge** | | | | | |
| 4AT | 3.1 (2.7) | 3.9 (3.4) | −0.45 | (−1.23 to 0.33) | 0.255 |
| HowRThey | 3.3 (2.8) | 2.5 (2.8) | 0.52 | (−0.65 to 1.69) | 0.387 |
| HowRwe | 8.9 (2.5) | 9.1 (2.4) | −0.35 | (−1.15 to 0.44) | 0.387 |
| Length of stay | 18.8 (10.2) | 16.6 (12.0) | 2.15 | (−0.70 to 5.01) | 0.139 |
| PERFECTER | 0.75 (0.11) | 0.74 (0.17) | 0.059 | (−0.10 to 0.21) | 0.450 |
| **1 month** | | | | | |
| BADLS | 25.0 (12.5) | 24.8 (13.6) | −1.50 | (−4.56 to 1.57) | 0.338 |
| EQ-5D SI | 0.8 (0.2) | 0.9 (0.2) | −0.029 | (−0.066 to 0.007) | 0.113 |
| EQ-5D by Proxy | 0.2 (0.3) | 0.3 (0.3) | 0.028 | (−0.042 to 0.099) | 0.434 |
| EQ-5D Patient | 0.6 (0.3) | 0.5 (0.4) | 0.074 | (−0.078 to 0.225) | 0.341 |
| HowRThey | 4.8 (2.6) | 4.0 (2.8) | 0.601 | (−0.040 to 1.241) | 0.066 |
| MMSE | 13.9 (8.0) | 13.0 (7.9) | 0.29 | (−1.04 to 1.62) | 0.669 |
| **3 months** | | | | | |
| BADLS | 24.6 (13.6) | 22.4 (13.4) | −0.46 | (−4.35 to 3.42) | 0.815 |
| EQ-5D SI | 0.8 (0.2) | 0.9 (0.2) | −0.017 | (−0.073 to 0.039) | 0.556 |
| EQ-5D Proxy | 0.3 (0.3) | 0.3 (0.3) | 0.071 | (0.018 to 0.124) | 0.009 |
| EQ-5D Patient | 0.6 (0.3) | 0.6 (0.4) | 0.024 | (−0.052 to 0.101) | 0.533 |
| HowRThey | 4.3 (2.5) | 3.4 (2.9) | 0.47 | (−0.53 to 1.47) | 0.359 |
| MMSE | 13.6 (8.6) | 12.5 (8.9) | 0.75 | (−0.77 to 2.27) | 0.333 |
| Timed Up and Go | 47.3 (33.3) | 48.7 (28.1) | −1.54 | (−15.38 to 12.30) | 0.827 |
| **6 months** | | | | | |
| BADLS | 26.4 (14.2) | 21.6 (12.0) | 1.97 | (−1.31 to 5.25) | 0.239 |
| CDR Score (SI) | 1.9 (1.1) | 1.7 (1.0) | −0.015 | (−0.160 to 0.131) | 0.845 |
| EQ-5D SI | 0.8 (0.2) | 0.9 (0.2) | −0.016 | (−0.096 to 0.063) | 0.688 |
| EQ-5D by Proxy | 0.4 (0.3) | 0.3 (0.4) | 0.099 | (0.001 to 0.198) | 0.047 |
| EQ-5D Patient | 0.7 (0.3) | 0.7 (0.3) | 0.057 | (−0.104 to 0.218) | 0.489 |
| HowRThey | 4.1 (2.7) | 3.3 (2.7) | 0.38 | (−0.49 to 1.25) | 0.394 |
| MMSE | 13.1 (9.3) | 12.2 (8.9) | 0.69 | (−1.14 to 2.53) | 0.457 |

*a: Estimated from a general linear model using generalised estimating equations. This model includes the baseline value of the modelled outcome where available.
BADLS, Bristol Activities of Daily Living Score; CDR, Clinical Dementia Rating.

**Table 4** Mortality and discharge destination outcomes

| Mortality | Intervention (N=132) (%) | Control (N=150) (%) | Total (N=282) (%) |
|---|---|---|---|
| Death in hospital* | 4 (4.0) | 7 (5.7) | 11 (4.9) |
| Death within 30 days of surgery† | 8 (6.1) | 9 (6.1) | 17 (6.1) |
| Death within 6 months of surgery† | 28 (21.4) | 24 (16.2) | 52 (18.4) |
| Total deaths | 30 (22.7) | 27 (18.0) | 57 (20.2) |
| NHFD discharge destination‡ | | | |
| Died | 4 (4.0) | 7 (5.7) | 11 (4.9) |
| Nursing care | 19 (19.0) | 16 (13.0) | 35 (15.7) |
| Other | 3 (3.0) | 1 (0.8) | 4 (1.8) |
| Own home/sheltered housing | 36 (36.0) | 58 (47.2) | 94 (42.2) |
| Rehabilitation unit (NHS-funded care home bed) | ‡ | 8 (6.5) | 8 (3.6) |
| Rehabilitation unit (hospital bed in another trust) | 12 (12.0) | 8 (6.5) | 20 (9.0) |
| Residential care | 21 (21.0) | 25 (20.3) | 46 (20.6) |
| Unknown | 5 (5.0) | ‡ | 5 (2.2) |
| Missing | 32 (24.2) | 27 (18.0) | 59 (20.9) |

*From NHFD data, not available for 59 Scottish participants, 32 intervention and 27 control.
†Three patients (one intervention, two control) included in 'total deaths' had missing surgery dates. These have not been included in the 'death within 30 days of surgery' or the 'death within 6 months of surgery' totals.
‡From NHFD data, not available for 59 Scottish participants, 32 intervention and 27 control.
NHFD, National Hip Fracture Database.

## Clinical outcome feasibility
The baseline characteristics and outcomes are presented in tables 3 and 4.

## Mortality
Over the duration of the trial, 57 participants (20%) died. A slightly higher rate was observed in the intervention group than in the control group, (23% vs 18%). Death in hospital was determined from NHFD data and only available for participants in England, thus excluding 59 Scottish participants. Eleven participants (5% of those with NHFD data) died in hospital with more in the control group (6% vs 4%). There were 17 (6%) patients who died within 30 days of surgery and 52 (18.4%) within 6 months.

## Discharge destination
Place of discharge from hospital was identified from the NHFD data, thus unavailable for 59 Scottish participants. The largest proportion of participants returned to their own home or moved into sheltered housing (42%). This destination was more likely in the control group (47%) than the intervention group (36%).

## Quality of life
No differences were seen in health-related quality of life (HRQOL) between the control group and intervention group at discharge or 1-month follow-up. At 3 months, a

potential beneficial effect of the intervention over control was evidenced for patient HRQOL based on the EQ-5D-5L by proxy: those in the intervention group had a mean EQ-5D utility score 0.071 higher than control (95% CIs: (0.018 to 0.124), p=0.009), a relatively small effect of around one quarter of an SD. A difference of 0.099, in favour of the intervention group, was also seen at the 6 months follow-up (95% CIs: (0.001 to 0.198), p=0.047).

## Economic outcome feasibility
Intervention costs across the five study wards ranged from £131 to £485 per patient over the study period (online supplemental table 5). There were no significant differences in total costs between groups at any time point except in total health and social care (HSC) costs (including intervention costs) at 3 months using suitable informant reported data (£4004, 95% CIs: £30 to £7979, p=0.049). Total costs (including intervention costs) at each time point are summarised in online supplemental table 6.

Total costs over the intervention period (online supplemental table 7) differed depending on the perspective and the source of data on hospital utilisation. HSC costs based on suitable-informant-reported data, including or excluding intervention costs, were significantly higher in the intervention than control group. However groups did not differ on total societal costs, including or excluding intervention costs, regardless of source. Suitable informant data differed from the hospital records-extracted data in that it could include hospital stays from trusts other thans those providing the hospital records, which may partly explain discrepancies between costs from different sources.

Group ICCs for 6 months costs and QALY are given in online supplemental table 8 . In the costs data, a pattern of negative ICC estimates indicated little clustering in the intervention group but some degree of clustering in the control group data. ICC for QALY ranged from 0.004 to 0.268 in the intervention and from −0.04 to 0.263 in the control group.

Concordance between hospital records-extracted and suitable-informant-reported sources on frequency of hospital service use and costs was generally weak, although Lin coefficients ranged between $\rho_c = 0.099$ and $\rho_c = 0.813$ for service use across time points (online supplemental table 9). Concordance on hospital costs was high at the baseline ($\rho_c = 0.660$) but was $\rho_c = 0.379$ at 1 month and $\rho_c < 0.3$ at three and 6 months. Limits of agreement showed that the two measures yielded estimates within £3400 of each other at baseline, £7000 at 1 month and similar at 6 months, but at 3 months the limits of agreement were much wider (£8020 to £10 693).

## Sample size calculation
ICCs were estimated, with 95% CIs to inform a sample size calculation. The highest value was estimated for the PERFECT-ER score, 0.748, indicating a substantial degree of between-hospital variation compared with variation between-individuals within hospitals. This is not surprising given the intervention aimed to standardise practice within intervention hospitals thereby inflating the ICC. At follow-up time points, the ICCs typically ranged between 0.05 and 0.1. At

6months, estimates for the MMSE and EQ-5D-5L by proxy were negative and, since a negative value is theoretically not possible and results from estimation error, these were interpreted as being a 'small', positive value, near to zero.

## DISCUSSION

The findings indicate that modifications are necessary to the trial design for a viable definitive trial. While this study successfully demonstrated the ability to recruit from a variety of different UK sites, the rate was lower than anticipated. There was a lot of missing data for some measures, therefore, steps to improve retention of participants at follow-up time points is warranted, and a sufficiently large inflation of the sample size is required to compensate for missingness. Mortality has been suggested as an appropriate primary outcome. Economic data collection proved burdensome to suitable informants. A definitive trial should reduce this burden for example, by extracting hospital services use data from hospital records.

We hypothesise short-term mortality (30 days) may be reduced by the PERFECT-ER intervention due to the cumulative effect of increased good practices across the range of care domains. This builds on previous work[10 36–38] which recognises complex associations between hospitalisation, pre-admission CI, postadmission CI, functional decline and mortality. Through this, we would recommend mortality be a proposed primary outcome if a future definitive trial is undertaken.

Complex interventions that focus on staff quality improvement and associated implementation methods such as plan–do–study–act methods[22] present challenges for investigation using RCTs.[39] The management and care of people with dementia and CI with hip fracture is complex. This is an example of a 'wicked problem', defined as complex, messy and stubborn challenges which continually evolve and has, at its core, many reasons for being, with no single solution which can be applied in all circumstances. Ultimately 'wicked problems' are those which cannot be reduced to a set of fixable problems and are often impossible to 'solve' because of incomplete, competing and changing requirements and where the solutions needed are 'better or worse' rather than 'right or wrong'.[40–42] While pragmatic RCTs, which offer tailoring and flexibility in experimental interventions, are one approach to testing management strategies for such healthcare challenges, other research methodologies may provide important insights. Further consideration of a range of methodological approaches may be more appropriate to answer this research question before automatically embarking on a clinical trial pathway.

## CONCLUSION

This study has demonstrated that PERFECT-ER can be implemented and widely accepted across a number of different health services in the UK's NHS. We have shown it is feasible, with modifications, to undertake a definitive trial and economic evaluation using the developed and refined recruitment and consenting practices. However, care of people with CI and hip fracture poses a 'wicked problem' and further definitive research using an RCT approach should be deliberated against other methods of evaluation.

**Author affiliations**
[1]School of Health Sciences, University of East Anglia, Norwich, UK
[2]School of Education and Lifelong Learning, University of East Anglia, Norwich, UK
[3]Norwich Medical School, University of East Anglia, Norwich, UK
[4]Care Policy and Evaluation Centre (CPEC), London School of Economics, London, UK, London, UK
[5]Norfolk and Norwich University Hospital, Norwich, UK
[6]Geriatric Medicine Unit, University of Edinburgh Western General Hospital, Edinburgh, UK
[7]HCOP, Nottingham University Hospitals NHS Trust, Nottingham, UK
[8]University of Birmingham, Birmingham, UK
[9]King's College London, London, UK
[10]College of Medicine and Health, University of Exeter, Exeter, UK

**Contributors** JLC, SPH, LS, FP, CH, TB, BP, SD, DL, MK, AM, MP, OS, TOS, JW, RH, CB and CF made substantial contributions to the conception or design of the work; DL provided PPI input throughout the study. SPH and TB led on the acquisition of the data, LS, CF, SPH and JLC led the statistical analysis and interpretation of data for the work; JLC led the drafting of the paper. All authors were involved in revising it critically for important intellectual content; All authors reviewed the paper and gave their final approval of the version to be published; all authors give their agreement to be accountable for all aspects of the work in ensuring that questions related to the accuracy or integrity of any part of the work are appropriately investigated and resolved. CF accepts full responsibility for the work and/or the conduct of the study, had access to the data, and controlled the decision to publish.

**Funding** This work was funded by NIHR Programme Grants for Applied Research (PGfAR) Programme grant number (ref: DTC-RP-PG-0311–12004). the views expressed are those of the authors and not necessarily those of the NIHR or the Department of Health and Social Care

**Competing interests** None declared.

**Patient consent for publication** Not applicable.

**Ethics approval** Ethical approval for the trial was given by Camden and Kings Research Ethics Committee (reference number: 16/LO/0621) and Scotland Research Ethics Committee A (reference number: 16/SS/0086). Participants gave informed consent to participate in the study before taking part.

**Provenance and peer review** Not commissioned; externally peer reviewed.

**Data availability statement** All data relevant to the study are included in the article or uploaded as online supplemental information. Data requests should be directed to: Professor Chris Fox Clinical Professor, University of Exeter lhttps://orcid.org/0000-0001-9480-5704Email:Christopher.Fox@exeter.ac.uk.

**ORCID iDs**
Jane L Cross http://orcid.org/0000-0002-7003-1916
Simon P Hammond http://orcid.org/0000-0002-0473-3610
Tamara Backhouse http://orcid.org/0000-0001-8194-4174

Opinder Sahota http://orcid.org/0000-0003-0055-7637

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
