## [Reviewer comments · BMJ Open]

ARTICLE DETAILS

TITLE (PROVISIONAL)	PERFECTED enhanced recovery pathway (PERFECT-ER) versus standard acute hospital care for people after hip fracture surgery who have cognitive impairment: a feasibility cluster randomised controlled trial
AUTHORS	Cross, Jane; Hammond, Simon; Shepstone, Lee; Poland, Fiona; Henderson, Catherine; Backhouse, Tamara; Penhale, Bridget; Donell, Simon; Knapp, Martin; Lewins, Douglas; MacLulich, Alasdair; Patel, Martyn; Sahota, Opinder; Smith, Toby; Waring, Justin; Howard, Robert; Ballard, Clive; Fox, Chris

VERSION 1 – REVIEW

REVIEWER	Lim, Jae-Young Seoul National University Bundang Hospital, Rehabilitation Medicine
REVIEW RETURNED	15-Aug-2021

GENERAL COMMENTS	Dear authors, I'm happy to have an opportunity for reviewing your study. The authors assessed the feasibility of a cluster RCT with economic analysis to measure the clinical and cost-effectiveness of an Enhanced Recovery Pathway for people with hip fracture and Cognitive Impairment. I agree the care and recovery of older patients with cognitive impairment remains an ongoing area of concern in fragility fracture care. I think this study is generally well designed and the manuscript is well written. However, some issue needs to be clarified. In this study, the experimental intervention may not be described sufficiently. I wonder how PERFECT-ER was specifically implemented in acute hospital care for people after hip fracture surgery. The experimental procedure needs to be explained more in detail. In addition, little is explained about the usual care given to the control group. Please, offer a brief outline of what the usual care is. In this feasibility study, it was not clearly stated what should be modified and to what extent for a viable definitive trial. The study limitations need to be discussed adequately. Thanks.
--

REVIEWER	Neuman, Mark University of Pennsylvania
REVIEW RETURNED	27-Oct-2021

GENERAL COMMENTS	This is a well-written and clearly presented summary of an innovative pilot RCT to evaluate a care improvement intervention for persons with hip fracture and cognitive impairment. High-quality work on this topic is much needed for reasons pointed out by the authors in the introduction section. As such, it is likely to be impactful and the results from the companion process paper and follow-on trials will be read with great interest by several communities interested in improving care for this group. I have only a few minor comments on presentation of the work. While it is acknowledged that the authors will be presenting a separate analysis of their implementation process that may contain more information on specific intervention components, it would still be helpful to have some information on this topic in the present article for the purposes of framing. The authors are encouraged to consider a table or figure describing the principal components of the intervention. As the authors note, the high rate of missing data at baseline is concerning. Can the authors comment on potential reasons for this? Presumably, this may have occurred due to competing clinical care processes early on during hospitalization for hip fracture. If this is the case, the solutions proposed to this problem by the authors do not seem optimal, since clinical care needs may still override data collection as a priority for many patients. Inflating sample size also does not seem to be a solution, since generalizability of results will still be problematic. One question for the authors to consider is whether the instrument battery at baseline could be made more parsimonious. What instruments are absolutely critical to collect? Acknowledging the challenges involved in data collection in this context, could it still be possible to design a baseline assessment battery that could be done in just a few minutes? Given the intent to scale this work to a large number of hospitals, strategically reducing the number of assessments involved could potentially benefit feasibility. The 24% figure quoted in the introduction (reference 4) seems implausibly high and likely needs to be contextualized. I am not sure that I am convinced by the authors' statement regarding the relevance of RCTs in the present context. While it is acknowledged that RCTs in this setting will face serious challenges, the insights gained from such work would be distinct and complementary in important ways to other methods they mention such as case study. In other words, it is not clear if this is really an either/or choice vs a both/and decision that would need to incorporate decisions of timing and structure into when a trial should be pursued. A reasonable finding from this work may be that more study is still required before proceeding to a rigorous test of different models of care, though I do not take the results here to argue against the value of trials as a mode of inference in this case. The term "wicked problem" feels pejorative. Can the authors find an alternate wording?
---

VERSION 1 – AUTHOR RESPONSE

In this study, the experimental intervention may not be described sufficiently. I wonder how PERFECT-ER was specifically implemented in acute hospital care for people after hip fracture surgery. The experimental procedure needs to be explained more in detail.

The implementation process and findings from this aspect of the research are submitted elsewhere awaiting review. The experimental procedures are fully detailed in the published protocol and referenced in this paper

In addition, little is explained about the usual care given to the control group. Please, offer a brief outline of what the usual care is.

Added control groups followed NICE guidance for hip fracture care with a reference

In this feasibility study, it was not clearly stated what should be modified and to what extent for a viable definitive trial.

Lessons from the feasibility study are reported, a primary outcome measure is suggested, and modifications to other outcome measures are discussed.

The study limitations need to be discussed adequately.

Study limitations are described in the strengths and limitations section and again in the discussion

consider a table or figure describing the principal components of the intervention.

The implementation process and findings from this aspect of the research are submitted elsewhere awaiting review. The experimental procedures are fully detailed in the published protocol and referenced in this paper

As the authors note, the high rate of missing data at baseline is concerning. Can the authors comment on potential reasons for this? Presumably, this may have occurred due to competing clinical care processes early on during hospitalization for hip fracture. If this is the case, the solutions proposed to this problem by the authors do not seem optimal, since clinical care needs may still override data collection as a priority for many patients. Inflating sample size also does not seem to be a solution, since generalizability of results will still be problematic. One question for the authors to consider is whether the instrument battery at baseline could be made more parsimonious. What instruments are absolutely critical to collect? Acknowledging the challenges involved in data collection in this context, could it still be possible to design a baseline assessment battery that could be done in just a few minutes? Given the intent to scale this work to a large number of hospitals, strategically reducing the number of assessments involved could potentially benefit feasibility.

As a feasibility trial – we report the broad range of data collection tools and outcome measures which were subject to testing. The intention would therefore be to reduce these significantly in any definitive trial after defining the primary outcome. Additional data collection methods, particularly for the health economics, would be subject to significant review. However, it is the purpose of this paper to report the findings of the feasibility and the lessons learnt as discussed in this paper

The 24% figure quoted in the introduction (reference 4) seems implausibly high and likely needs to be contextualized.

Thank you for raising this - there was a typo/ editing error here and the figure has been amended to 15% after checking the reference.

I am not sure that I am convinced by the authors' statement regarding the relevance of RCTs in the present context. While it is acknowledged that RCTs in this setting will face serious challenges, the insights gained from such work would be distinct and complementary in important ways to other methods they mention such as case study. In other words, it is not clear if this is really and either/or choice vs a both/and decision that would need to incorporate decisions of timing and structure into when a trial should be pursued. A reasonable finding from this work may be that more study is still required before proceeding to a rigorous test of different models of care, though I do not take the results here to argue against the value of trials as a mode of inference in this case.

It is not the intention of this paper to argue against the value of trials the authors are proposing that a variety of methods are needed to investigate this research problem in a meaningful way and that a single trial is perhaps not the answer – instead we propose the need to examine the practice challenges for interventions and their implementation in the context of ongoing practice.

The term "wicked problem" feels pejorative. Can the authors find an alternate wording?

wicked problems" is a term used in service and policy research literature to identify issues which are not simply 'fixable' problems – (see refs 41-43). To further clarify this term, we have modified the two sentences following our use of this term have been modified.

VERSION 2 – REVIEW

REVIEWER	Lim, Jae-Young Seoul National University Bundang Hospital, Rehabilitation Medicine
REVIEW RETURNED	14-Jan-2022
GENERAL COMMENTS	I appreciate the authors' efforts to correct the manuscript as suggested. The quality of the manuscript has improved considerably in the revised form. Thanks.